# Differences in the Abundance of Auxin Homeostasis Proteins Suggest Their Central Roles for In Vitro Tissue Differentiation in *Coffea arabica*

**DOI:** 10.3390/plants10122607

**Published:** 2021-11-27

**Authors:** Ana O. Quintana-Escobar, Hugo A. Méndez-Hernández, Rosa M. Galaz-Ávalos, José M. Elizalde-Contreras, Francisco A. Reyes-Soria, Victor Aguilar-Hernández, Eliel Ruíz-May, Víctor M. Loyola-Vargas

**Affiliations:** 1Centro de Investigación Científica de Yucatán, Unidad de Bioquímica y Biología Molecular de Plantas, Calle 43, No. 130 x 32 y 34, CP, Mérida 97205, Mexico; ana.quintana@estudiantes.cicy.mx (A.O.Q.-E.); hugo.mendez@cicy.mx (H.A.M.-H.); gaar@cicy.mx (R.M.G.-Á.); 2Instituto de Ecología A.C. (INECOL), Red de Estudios Moleculares Avanzados, Clúster Científico y Tecnológico BioMimic^®^, Carretera Antigua a Coatepec No. 351, Congregación el Haya, CP, Xalapa 91070, Mexico; jose.elizalde@inecol.mx (J.M.E.-C.); antonio.ry.16@gmail.com (F.A.R.-S.); 3Centro de Investigación Científica de Yucatán, Catedrático CONACYT, Unidad de Bioquímica y Biología Molecular de Plantas, Mérida 97205, Mexico; victor.aguilar@cicy.mx

**Keywords:** *Coffea arabica*, cellular differentiation, mass spectrometry analysis, plant tissue culture, quantitative proteomics, tandem mass tag

## Abstract

*Coffea arabica* is one of the most important crops worldwide. In vitro culture is an alternative for achieving *Coffea* regeneration, propagation, conservation, genetic improvement, and genome editing. The aim of this work was to identify proteins involved in auxin homeostasis by isobaric tandem mass tag (TMT) and the synchronous precursor selection (SPS)-based MS3 technology on the Orbitrap Fusion™ Tribrid mass spectrometer™ in three types of biological materials corresponding to *C. arabica*: plantlet leaves, calli, and suspension cultures. Proteins included in the β-oxidation of indole butyric acid and in the signaling, transport, and conjugation of indole-3-acetic acid were identified, such as the indole butyric response (IBR), the auxin binding protein (ABP), the ATP-binding cassette transporters (ABC), the Gretchen-Hagen 3 proteins (GH3), and the indole-3-acetic-leucine-resistant proteins (ILR). A more significant accumulation of proteins involved in auxin homeostasis was found in the suspension cultures vs. the plantlet, followed by callus vs. plantlet and suspension culture vs. callus, suggesting important roles of these proteins in the cell differentiation process.

## 1. Introduction

Coffee is one of the most important crops worldwide. The genus *Coffea* is composed of more than 130 species, of which *Coffea arabica* and *C. canephora* are the most economically important [1], as they represent around 60% and 40% of world coffee production, respectively [2]. There is a high demand for *Coffea* spp. plants production to meet the growing demand for coffee in the market. These plants must be of high quality and resistant to a range of diseases that affect the genus. Vegetative propagation is the preferred method for the large-scale production of superior plants [3] to maintain the desired characteristics of the mother plants. This type of propagation also allows the development of high homogeneous resistance to pests and diseases in a short time and limited space [4]. Therefore, plant tissue culture has long been a preferred method for the conservation [5], propagation, and genetic improvement of recalcitrant cultures.

Somatic embryogenesis (SE) has been positioned as an effective tool for propagation compared to conventional methods, either by seed or by cuttings [6], for both commercial or research purposes. SE can be achieved directly on the explant or indirectly through callus. The first route is of low frequency since the number of embryos obtained is lower, while the second one is of much higher frequency and is preferred for achieving mass propagation [7]. Currently, SE is a useful biotechnological tool for propagation, genetic transformation, and genetic improvement, as well as for basic research on the molecular mechanisms underlying SE [8]. The study of SE in coffee has been carried out since 1970 [9]; the two main foci of research since then have been to improve the methodology and to understand the mechanism by which somatic embryos are obtained [10].

Because SE in coffee can be started from different tissues, such as suspension cultures [11,12,13], calli [14], leaves [15], and others [10], one critical element to understand is the SE induction mechanism. Suspension cultures are an effective substrate for metabolism research since they can be synchronized [16]. Calli is a group of dedifferentiated and disorganized cells, useful for genetic transformation studies [16]. Differentiated organs, such as leaves, contain different types of cells and are preferred for starting SE due to their availability and also because the embryos are obtained faster, although in smaller numbers [1,3,17]. Leaves are preferred because they are a more viable and available source of clonal explants from elite selected genotypes [18,19], and within the culture medium, they have a greater contact area with nutrients in the zone of the explant wound. In addition, SE origin can be unicellular or multicellular. Periclinal cell division has been observed to initiate rapidly in cells adjacent to the leaf surface and to a lesser extent in epidermal or parenchymal cells [20].

Plant growth regulators play an essential role in all aspects of plant growth and development [21], of which auxins are one of the most important groups. Auxins have a primary role in cell division, elongation, differentiation, organogenesis, embryogenesis, and response to external stimuli, as well as in the formation of cells and tissues [22]. For those processes to be carried out, regulation of biosynthesis, conjugation, transport, and signaling is required, which integrates auxin homeostasis [23]. Biological processes such as directional transport and the formation of auxin gradients are achieved by the different input transporters such as the AUXIN RESISTANT1/LIKE AUX1 (AUX1/LAX) [24], and output transporters such as the PINs [25] and the ABCB (ATP-BINDING CASSETTE subfamily B) [26]. Regarding their conjugation, several genes that code for indole-3-acetic acid (IAA) amido synthetases have been identified, such as those belonging to group II of the *GH**3* family [27].

Different conjugates of IAA have been implicated in various biological processes; for example, IAA-Glu is synthesized during induction of SE in *Coffea canephora* and is considered a precursor of auxin degradation [28]. However, conjugates such as IAA-Leu and IAA-Ala can be hydrolyzed to release free IAA through amido hydrolase enzymes encoded by genes of the ILR1 family [29].

The use of omics sciences has been crucial for the study of plant development in recent years [1]. With modern omics tools, it is possible to understand the molecular mechanism that leads to the formation of embryos, starting from somatic cells. Proteomics is a valuable tool for studying protein levels during plant development [16] and somatic embryogenesis [30]. Various proteomic studies on differentiation in *Coffea* species have been carried out using different types of initial explants, such as suspension cultures [11], calli [6], and leaves [31]. In proteomic techniques, the differential labeling of peptides with isobaric tags, such as the tandem mass tag (TMT), reduces handling and analysis time, allowing the quantification of peptides by measuring the intensity of the reporter ion [32]. Furthermore, the application of synchronous precursor selection (SPS)-MS^3^ technology, available as a hybrid platform in the Orbitrap Fusion^TM^ Tribid, provides the means of eliminating contaminants by isolating near-isobaric ions that fragment together with the target ions [33]. This was the approach used in this study, as it yields robust comparative proteomics data without ratio distortion in isobaric multiplexed quantitative proteomics. Likewise, proteomic analysis allows the identification of molecular markers associated with in vitro morphogenesis [34]. This work aims to identify which proteins involved in auxin homeostasis may be involved in the process of cell differentiation by comparing calli, suspension cultures, and plantlet leaves from *C. arabica*.

## 2. Results

Three types of plant tissues grown in vitro were used to identify the differentially abundant proteins between: calli vs. plantlet leaves (CvsP comparison), suspension culture vs. plantlet leaves (SvsP comparison), and suspension culture vs. calli (SvsC comparison) (Figure 1A). In the 1D-SDS-PAGE, it is possible to observe the banding pattern of the total protein extract of each tissue, which highlights notable differences between them (Figure 1B). A total of 2614 proteins were identified among the three comparisons (Appendix A).

The heat map (Figure 2) shows the difference between the distributions of the protein abundances of each tissue. Differences in the distribution of proteins were found. Two clusters were formed under the tissue comparisons. The accumulation of proteins in the suspension cultures was more similar to that in the calli than that in the plantlets. In contrast, the distribution of the proteins in the SvsC comparison had a starker contrast than the other two. There are slightly more down-accumulated proteins (green) than up-accumulated (red) when comparing suspension cultures against calli. On the other hand, more up- than down-accumulated proteins are seen when suspensions and calli are compared against plantlets.

The total of differentially accumulated proteins (DAPs) was 744, 982, and 295 for the CvsP, SvsP, and SvsC comparisons, respectively (Figure 3). Of these, the number of highly accumulated proteins (up-accumulated) was 414, 541, and 120, respectively, while the number of proteins less accumulated (down-accumulated) was 330, 441, and 175 (Figure 3A). From the total proteins, the separation of those differentially accumulated was carried out according to their fold change in relative abundance (up >1.5; down <0.66; *p* < 0.05). In accordance with what was observed from the global protein abundance presented in Figure 2, in the CvsP and SvsP comparisons, the number of up-accumulated proteins was higher than those down-accumulated. In contrast, in the SvsC comparison, the opposite occurred, where there were a higher number of down-accumulated proteins (Figure 3A). When comparing SvsC, the lowest number of DAPs was found, suggesting that both tissues are at a similar level of differentiation. Therefore, significant changes at the proteomic level would not be expected. On the contrary, more DAPs were found in the SvsP comparison, as they are remarkably distinct at the differentiation stage.

A Venn diagram was generated to visualize specific and shared DAPs between the different sets of samples (Figure 3B). The highest number of unique DAPs was found in the SvsP comparison, followed by CvsP and SvsC, with 257, 60, and 36 proteins, respectively. Out of the 257 DAPs in SvsP, 159 were up-, and 98 were down-accumulated, and out of the 60 DAPs in CvsP, 50 were up-and 10 down-accumulated. On the other hand, with only 36 unique proteins, the SvsC comparison presents the lowest number of unique DAPs, of which 24 were up- and 12 down-accumulated (Figure 3B).

A gene ontology analysis for each tissue comparison was performed to classify up-accumulated DAPs according to the biological processes, molecular functions, and cellular components. In the CvsP comparison (Figure 4A), the most enriched biological processes were the catabolic process, the biosynthetic process, carbohydrate and nitrogen metabolism, response to stress, precursor metabolites, and energy. The most enriched molecular functions were ion binding, oxidoreductase, hydrolase and kinase activity, and transmembrane transporter activity. The cytosol, plasma membrane, extracellular region, plastid, and mitochondrion were the most enriched cellular components. According to the hierarchical grouping, the most significant routes correspond to the response to toxic substances and antioxidant activity (Figure 4B).

In addition to biological processes found in CvsP, the SvsP comparison included translation, and cellular component organization, among others (Figure 5A). The most enriched cellular components were the cytosol, nucleus, mitochondrion, ribosome, and endomembrane system. Some of the previous molecular functions were also enriched, in addition to those associated with protein binding. The most notable routes correspond to those involved in peptide metabolism (Figure 5B).

In the SvsC comparison (Figure 6), the most enriched biological processes were mRNA processing, ribosome biogenesis, protein folding, and the biosynthetic process, among others. The most enriched molecular functions were mRNA and ion binding, and some belonging to protein metabolism. The protein-containing complex, cytosol, plastid, Golgi apparatus, and ribosome were the most enriched cellular components. In this comparison, it was impossible to identify the most significant routes to perform the hierarchical grouping due to the small number of identified DAPs.

In addition, we identified 126 DAPs shared among the three comparisons (Figure 3B). These proteins are involved in essential functions, such as the metabolism of energy, carbohydrates, lipids, amino acids, and biosynthesis of other secondary metabolites. In addition, a hierarchical GO enrichment grouping was carried out (Figure 7). The correlation between the functional categories of the significantly enriched routes of the 126 proteins found to accumulate continuously is summarized. The functions related to photosynthesis and energy generation were the most active.

A manual search was carried out for the proteins involved in auxin homeostasis. Members of families responsible for signaling, transport, conjugation, hydrolysis, and β-oxidation were found, such as ABP, ABC, BIG, GH3, ILR, IBR, and UGT. GH3 proteins play a crucial role in auxin homeostasis through the conjugation of IAA with various amino acids [35]. For example, conjugates such as IAA-Asp and IAA-Glu are considered precursors of an irreversible degradation pathway for IAA [36]. We found a pair of GH3.17 proteins (CaGH3.17a and CaGH3.17 b) within our *C. arabica* proteome. In *A. thaliana*, AtGH3.17 has been reported to correspond to group II. This group is involved in the conjugation of auxin with amino acids [37].

We performed a BLAST analysis with our GH3.17a and GH3.17b sequences against The Arabidopsis Information Resource database (TAIR). The sequence GH3.17a from *C. arabica* shared 79% identity and 91% similarity with the GH3.17 protein from Arabidopsis (Appendix A), while the sequence GH3.17b shared 61% identity and 75% similarity with the Arabidopsis GH3.17 protein (Appendix A). We subsequently analyzed 47 GH3 protein sequences to build a phylogenetic tree (Appendix A) using the GH3 proteins of *O. sativa* as an outer group (Figure 8). In the phylogenetic tree, the GH3 proteins were grouped into four clades. We observed the three groups previously reported in Arabidopsis (I, II, III). Group I proteins consist only of AtGH3.10 and AtGH3.11 [38]. Group II enzymes catalyze the formation of conjugates between auxins (mainly IAA) and amino acids, which function as a regulatory mechanism to maintain auxin homeostasis [37]. The CaGH3.17a and CaGH3.17b proteins from *C. arabica* clustered with the AtGH3.17 protein from *A. thaliana* (Figure 8) and possibly fulfill the same function as amido synthetases. They were also grouped with other group II proteins related to the conjugation of IAA with amino acids (Figure 8).

Endogenous conjugates such as IAA-Ala, IAA-Leu, IAA-Phe have been reported to appear to be biologically active. They probably provide an easily accessible temporary storage form of auxin [39]. A family of amidohydrolases hydrolyzes these conjugates [40]. Currently, in Arabidopsis, the ILR1-like family consists of seven members: ILR1, ILL1, ILL2, ILL3, IAR3, ILL5, and ILL6 [41]. The best-characterized are ILL1, ILL2, and IAR3 that show more significant catalytic activity with IAA-Ala, while ILR1 prefers IAA-Leu and IAA-Phe as substrates [29]. ILL3 and ILL6 show no activity on IAA conjugates in vitro [42].

We also identified the sequences of the proteins that participate in the hydrolysis of the auxin conjugates, corresponding to the proteins ILR1-like 1, ILR1-like 2, and ILR1-like 4. We carried out a BLAST analysis with each of our *C. arabica* ILR1 sequences against The Arabidopsis Information Resource database (TAIR). The CaILR1-like 1 sequence of *C. arabica* shared 76% identity and 90% similarity with the Arabidopsis AtILR1-like 1 protein (Appendix A), the CaILR2-like 2 sequence shared 55% identity and 73% of similarity with the Arabidopsis AtILL2 protein (Appendix A). In comparison, the *C. arabica* ILR1-like 4 sequence shared 73% similarity and 90% identity with the Arabidopsis AtIAR3 (ILL4) protein (Appendix A). We constructed a phylogenetic tree using 24 protein sequences considered amidohydrolases (Appendix A), using ILR proteins from *O. sativa* as an outer group (Figure 9). In the phylogenetic tree, the ILR proteins were grouped into four clades. The CaILR1-like 1 protein from *C. arabica* clusters with the AtILL1 and AtILR1 proteins from *A. thaliana*. On the other hand, the CaILR-like 2 protein from *C. arabica* clusters with the AtILL2 protein from *A. thaliana.* In *A. thaliana*, deficient *ilr1* mutants have been reported to show reduced sensitivity to root elongation caused by the exogenous application of auxin conjugates such as IAA-Leu [43]. While mutants that overexpress *ilr1* and *ill2* show greater sensitivity to root elongation caused by conjugates with nonpolar amino acids such as IAA-Ala and IAA-Phe [41]. CaILR1-like 4 is orthologous with AtIAR3 (ILL4) from Arabidopsis and is also in the same clade as the previous ones, possibly participating in the hydrolysis of conjugates to maintain auxin homeostasis and release it when the cell requires it.

In *A. thaliana*, the ABCB subfamily includes 21 members distributed in three clades: ABCB1, ABCB14, and ABCB19 in clade I, ABCB4 in clade II, and ABCB15 in clade III [44]. The ABCB1, ABCB4, and ABCB19 proteins have been well characterized as auxin transporters. However other ABC proteins such as ABCB14, ABCB15, and ABCB21 are linked to auxin transport [26,45,46]. In our work, we found eight proteins of the ABC family; however, we will focus the analysis on the members of the ABCB subfamily (ABCB2, ABCB4, and ABCB14). A BLASTp analysis of each of our sequences (ABCB2, ABCB4, and ABCB14) was carried out against TAIR to determine the possible homologs with *A. thaliana*. The ABCB2 sequence of *C. arabica* shared 78% identity and 90% similarity with the Arabidopsis ABCB2 protein. The *C. arabica* ABCB4 sequence shared 65% identity and 79% similarity with the Arabidopsis ABCB11 protein. In comparison, the *C. arabica* ABCB14 sequence shared 74% identity and 86% similarity with the Arabidopsis ABCB21 protein (Appendix A). We also performed a phylogenetic analysis of members of the ABCB subfamily, selecting a total of 31 protein sequences, the outer group being the ABCB proteins from *O. sativa*. The tree included the three groups described above (Figure 10).

Because this work aimed to analyze the proteins involved in IAA homeostasis, they were manually selected from the global proteome to compare abundance among the different tissues. One protein involved in IAA signaling was found, the auxin binding protein 20 (ABP20). Eight proteins belonging to the family of ABC transporters and an auxin transport protein, BIG, were also identified. Two proteins of the GH3 family responsible for conjugation were found, and three ILR1 amidohydrolases that participate in the conjugates’ hydrolysis. It was also possible to identify two proteins belonging to the indole butyric response (IBR) family involved in the β-oxidation of indole butyric acid (IBA) and one UGT protein, which is likely to be participating in the conjugation of IBA with sugars. Our findings provide an overview of auxin homeostasis in *C. arabica* and provide a solid basis for further experiments investigating the role of auxin homeostasis in regulating callus and suspension formation in *Coffea.*

A model was made with the set of identified proteins, which summarizes each of them in the different tissue comparisons (Figure 11).

Of the eight proteins identified from the ABC family, three were type B, one was type C, one was type E, two were type F, and one was type I, which remained in low abundance in the three comparisons. The ABP20 protein involved in IAA transport had a low accumulation in all three tissue comparisons; likewise, the accumulation of the auxin transport protein BIG remained unchanged in all three comparisons. Another five proteins had moderately high accumulation in all three tissue comparisons (ABCB.2, ABCC.2, ABCE.2, ABCF.3 and ABCF.5); meanwhile, the abundance of ABCB.4 and ABCB.14 was much lower in the SvsC comparison only. Of the two GH3.17 proteins, one had high accumulation compared to calli and suspension cultures against plantlets, while the other, GH3.17, remained in low abundance in all three comparisons. Three ILR1 were found, of which the ILR1-like 2 remained with low abundance in every comparison.

On the other hand, the ILR1-like 1 and 4 had similar behavior with a relatively high accumulation in all three comparisons. The IBR1 protein was highly abundant in the SvsP comparison, while the abundance of the IBR3 was lower in all comparisons. In addition, the UGT75C1 accumulation was very high in the SvsP comparison, while in the SvsC comparison, the accumulation was low. According to the model, it is inferred that the most significant changes in terms of the abundance of most of the proteins involved in auxin homeostasis occurred when the suspension cultures were compared with the plantlets (SvsP). On the contrary, fewer differences were observed when comparing suspension cultures with calli (SvsC).

## 3. Discussion

Given the worldwide economic relevance of *C. arabica*, there are many investigations with various approaches regarding its in vitro culture, even though it is a non-model plant. However, there is little information regarding the proteomic study in different types of tissues used.

There are several studies where different proteomic tools were used to answer a wide range of questions in *C. arabica*, using different types of plant materials: leaf [47,48], seeds [49], grains [50,51], suspension cultures [11,52], and somatic embryos [31]. In most of these studies, it has been possible to identify proteins related to stress, photosynthesis, energy generation, and carbohydrate metabolism, which is consistent with the results of the GO enrichment presented in our study.

It is not uncommon to find stress-related proteins since during in vitro tissue culture explants are subjected to various types of stress under certain conditions like frequent subcultures, changes in culture medium, changes in osmotic pressure, mechanical injuries, high relative humidity, low ventilation, modifications in the concentrations of growth regulators, among others [53,54]. Due to all these reasons, cells in tissues grown in vitro modify their molecular mechanisms to respond to such a situation and thus continue cell division and growth [53].

The study most similar to ours was the one carried out by Campos [11], where the proteome of suspension cultures of *C. arabica* was analyzed. We found many common proteins with the study mentioned above, such as the phosphoglycerate dehydrogenase involved in L-serine biosynthesis, which we identify as mainly accumulated in calli and suspension culture. Likewise, our results coincide in identifying proteins involved in energy production, which are necessary to carry out cell differentiation, such as glyceraldehyde-3-phosphate dehydrogenases, fructose bisphosphate aldolases, enolases, among others.

The plant cell wall serves as a dynamic physical barrier, consisting of interconnected layers that contain cellulose, hemicellulose, pectin, lignin, and proteins [55]. Several cell wall-related proteins were identified, such as pectinesterases, glycosyltransferases, expansins, polygalacturonases, cellulose synthase, callose synthase, and endochitinases. In addition, proteins associated with plant cell assembly and biogenesis were accumulated in the callus. Proteins with the role in the cleavage and build of polysaccharides, including two xyloglucan endotransglucosylase, two cellulose synthases, and a callose synthase, were found. A remarkable feature of callus is the up-accumulation of seven pectin methylesterase inhibitors (PMEI). These inhibitors can modulate the de-methyl esterification of homogalacturonan, inhibiting the pectin methylesterase. Biochemical studies have found methyl esterified pectin highly in callus and variation during the somatic embryo formation from callus [56,57].

Growth regulators are vital factors for plant development. Auxins, in particular, are involved in a large number of processes during tissue development, such as gene expression, cell division, cell elongation, and differentiation [58]. The responses are dependent on the concentration of auxin, which in turn depends on its homeostasis. Homeostasis is controlled by several mechanisms, such as auxin biosynthesis, degradation, transport, and conjugation [36].

For IAA signaling to be carried out, the participation of several groups of proteins is required, including the ABP family. Within this family, ABP1 is the most studied because it is involved in auxin perception and binding with high specificity and affinity and has an essential role in several processes such as cell division and cell expansion [59,60]. We found an ABP20, although its abundance was low in all three comparisons. An ABP20 protein was identified in *Prunus persicaria*, which has an auxin binding motif homologous to the ABP1 protein, but with different specificity [61,62]. ABP20 is involved in the perception of auxins, and it has superoxide dismutase (SOD) activity [62]. Subsequently, auxin transport is carried out by other large groups of proteins, such as ATP-binding cassette (ABC), PIN-FORMED (PIN), and BIG proteins.

The ABC proteins comprise one of the largest families of plant proteins and participate in the transport of various molecules across the membrane, such as mineral ions, lipids, peptides, metals, secondary metabolites, and growth regulators such as auxin [63,64,65]. They also have a primary role in the cellular detoxification mechanism [66]. This family of transporters in plants is divided into eight subfamilies: A, B, C, D, E, F, G, and I. However, even though it is a very pervasive family of transporters, studies on individual members are scarce [67].

Members of five different subfamilies were identified in this study: B, C, E, F, and I. Proteins ABCB4 and ABCB14 showed a notable difference in their abundance between the three different tissue comparisons. In heterologous systems, it was identified that ABCB4 from Arabidopsis participates as an exporter or importer of auxin, depending on its concentration [68]. On the other hand, it is known that the ABCB14 regulates stomatal activity in the face of changes in CO_2_ concentration by importing apoplastic malate [69]. The earlier research supports what was observed in this study, as the high accumulation of both proteins ABCB4 and ABCB14 was more significant in the comparisons against plantlets. In contrast, when comparing calli versus suspensions, the accumulation of ABCB4/14 was negative because both are tissues that lack chloroplasts, and, therefore, there are no stomatal cells to regulate.

ABC transporters involved in auxin transport have been characterized in mono and dicotyledonous species, suggesting a high level of conservation in the plant kingdom. In our homology analysis, the CaABCB4 protein from *C. arabica* showed similarity with the AtABCB11 protein from *A. thaliana*, and in the phylogenetic analysis, the CaABCB4 protein clusters in clade II. These results suggest that CaABCB4 from *C. arabica* may serve a similar role as a possible exit transporter and participate in cellular auxin homeostasis. Regarding the CaABCB14 protein, we observed that it shares a similarity with the AtABCB21 protein from *A. thaliana*. Both proteins were grouped in clade II of the phylogenetic tree. This phylogenetic tree is interesting, and possibly the sequence similarity implies a functional redundancy between these proteins. For example, it has been reported that AtABCB21 can function as an importer/exporter that controls auxin concentrations in the cell [70]. Recently, the transporter AtABCB21 has been reported to maintain the acropetal auxin transport. This transporter also regulates auxin levels in the pericycle and auxin distribution in the cotyledons and in the young leaves [26]. In our system, these proteins may help regulate the initial accumulations of auxin and are involved in different cell differentiation processes.

The ABCs of subfamily C are involved in the vacuolar transport of some compounds, as well as in the compartmentation of anthocyanins, detoxification, heavy metal sequestration, chlorophyll catabolite transport, and ion channel regulation [64,71]. The knowledge of this family of transporters is still limited in the processes of cellular differentiation. From the ABCC subfamily, we identified the CaABCC2 protein from *C. arabica*. However, its abundance was similar between the three comparisons. This protein has been reported to provide Arabidopsis resistance to heavy metals such as cadmium and arsenic, along with ABCC1 [72]. On the other hand, it has been reported that when the *Arabidopsis* AtABCC1 and AtABCC2 proteins are expressed heterologously, they exhibit transport activity of hormonal conjugates such as abscisic acid glucosyl ester (ABA-GE) [73].

Subfamily E is highly conserved in archaea, bacteria, and eukaryotes, which is why these family members are considered necessary for essential functions [67]. The proteins of the *A. thaliana* subfamilies ABCE and ABCF appear to consist of three and five domains, respectively, and probably participate in processes other than transport [66]. In this study, we identified the ABCE2 protein, which has been identified as participating in RNA silencing in Arabidopsis [74]. We also found two members of subfamily F, which is not yet well characterized in plants [64]; however, some studies suggest that ABCF3 and ABCF5 could be related to the response to stress. It has been demonstrated that ABCF3 is also involved in the control of protein translation, defense against pathogen infection, and regulation of H_2_O_2_ uptake by modulating the expression of aquaporin genes [75,76,77]. Subfamily I is found exclusively in plants [71], and they are involved in primary metabolism and responses to stress. It was recently determined that certain members of this subfamily, including ABCI21 found in our study, are involved in modulating cytokinin responses during seedling growth and development [78]. However, we identified that this protein is not abundant when comparing calli and suspensions against plants.

More than 90% of the auxin in plants is in conjugated form; that is, inactive [28,79]. When auxin levels are high, a more significant induction of *GRETCHEN-HAGEN 3* (GH3) genes is observed; these genes catalyze the formation of auxin-amino acid conjugates dependent on ATP [80]. Some records show an increased expression of *GH3.17* genes during cell differentiation [81]. Our homology analysis of the CaGH3.17a and CaGH3.17b proteins from *C. arabica* showed similarities with the AtGH3.17 protein from *A. thaliana*. They may act as amido synthetases to regulate auxin homeostasis during the cell differentiation process. In addition, the phylogenetic analysis groups the CaGH3.17a and CaGH3.17b proteins in group II that has the most studied members in auxin conjugation and includes the *A. thaliana* protein AtGH3.17. It has been suggested that in the presence of members of the *GH3* family, auxins can be conjugated to different amino acids [37]. The *GH3* family plays different roles during development. For example, in *A. thaliana*, the overexpression of the genes *AtGH3.2* and *AtGH3.6* belonging to group II showed a phenotype of short hypocotyls [82,83]. A *GH3.9* mutant, which also belongs to group II, showed a shorter root growth phenotype, indicating a role for *GH3s* in root development [84]. Despite being a multigene family, each *GH3* member may have specific roles in cell differentiation. On the other hand, some auxin conjugates (IAA–Ala, IAA–Leu, and IAA–Phe) can be hydrolyzed to return to their active form through the action of hydrolase enzymes such as ILR1. Conjugates with Asp and Glu belong to the degradation pathway [36]. ILR1s have been shown to reside in the endoplasmic reticulum, where hydrolases regulate the rates of amido-IAA hydrolysis, mainly of IAA-Leu and IAA-Phe, resulting in activation of the auxin signal and thus modulating auxin homeostasis [85]. In this study, it was determined that the CaILR1-like 1 protein of *C. arabica* is similar to the AtILR1-like 1 protein of *A. thaliana*. This suggested that it presents a similar function as amidohydrolase with an affinity for conjugates such as IAA-Leu and IAA-Phe, regulating the amount of free auxin when the cell requires it. This implies that the endoplasmic reticulum not only serves as an auxin storage compartment [86]. The CaILR1-like 2 and CaILR-like 4 proteins from *C. arabica*, which share similarities with the AtILL2 and AtIAR3 proteins from *A. thaliana*, were also identified. These results are interesting since it has been shown by in vitro enzymatic tests that the AtIAR3 and AtILL2 proteins show more significant catalytic activity using the AIA-Ala conjugate as substrate [29]. Auxins can also be conjugated to sugars using the uridine diphosphate (UDP) glycosyltransferases (UGTs) [36,87,88]. More than 100 UGT genes have been reported in *A. thaliana*, of which the majority encode functional enzymes [89] and can be organized into 14 groups according to their sequence similarity and evolutionary relationship [90]. In this study, UGT75C1 was identified as highly abundant in the SvsP comparison. Various UGTs have been characterized as participants in the control of the metabolism of different plant growth regulators [91]. However, there are only two reports of UGT75C1 found in plants: one in Arabidopsis [92] and another recently in *Lonicera japonica* [93], where it is thought to function as an anthocyanin-5-*O*-glucosyltransferase *in planta*. Nevertheless, more studies should be performed to confirm the role of UGTs in plant development.

In addition to the hydrolysis of the conjugates, another way to obtain free IAA is from IBA, through the elimination of two side-chain methylene in a β-oxidation process catalyzed by the indole butyric acid response enzyme (IBR) [94]. IBA is another naturally occurring auxin, structurally very similar to IAA, found either as a precursor or in storage form just like IAA conjugates [95]. The inactive forms of auxin help regulate homeostasis during development. In addition, IBA is widely used as an exogenous auxin in numerous species due to its effect as an inducer of root development [96].

The IBR1, IBR3, and IBR10 enzymes are implicated in IBA to IAA conversion [97]. In this study, the IBR1 and IBR3 proteins were found, of which IBR1 showed a high accumulation, mainly when comparing the suspensions with the plantlets. With studies conducted on Arabidopsis mutant seedlings, Strader [98] suggests that the β-oxidation of IBA through IBR1 generates the necessary auxin to carry out processes of cell differentiation and expansion. However, most of the research where the activity of IBR1 has been studied focuses on the root because it is an area with constant cell division. Based on our results and those mentioned above, we could hypothesize that the IAA transported and conjugated comes from the IBA β-oxidation pathway through the IBR1 enzyme.

The most common proteomic strategies used to study *C. arabica* have been 2-DE and MALDI-TOF; however, the liquid chromatography-mass spectrometry (LC-MS) tool has been gaining attention in recent years for proteomic studies because it offers a quantitative approach to the proteome [99]. Quantitative proteomics provides information on the molecular mechanisms that operate in the cell under various study conditions [100]. In this sense, the tandem mass tag allows a precise identification and quantification of proteins [101]. Few proteomic studies using gel-based techniques coupled with mass spectrometry have been performed in *C. arabica* suspension cultures and embryos [11,31,52], and, up to now, there are no reports of the use of TMT in in vitro tissue studies of this species so that this study can serve as a reference for the characterization of the proteome of *C. arabica* suspension cultures, calli, and plantlets.

## 4. Materials and Methods

### 4.1. In Vitro Tissue Growth Conditions

The samples used for proteomic analysis consist of in vitro plantlet leaves (P), calli (C), and suspension cultures (S) of *C. arabica*. The establishment and maintenance of the biological materials were carried out according to the methodology previously reported by Quiroz-Figueroa [102].

### 4.2. Protein Extraction

To extract the proteins, the plant tissue was triturated to obtain a fine powder, using a mortar and liquid nitrogen. The extraction buffer included 0.5 M Trizma base (pH 8; Sigma, T1503; St. Louis, MO, USA), 50 mM EDTA (pH 8; Sigma, EDS; St. Louis, MO, USA), 700 mM sucrose, 100 mM KCl (Sigma, P9541; St. Louis, MO, USA), 2% β-mercaptoethanol (Sigma, M6250; St. Louis, MO, USA), 1 mM PMSF (Sigma, 78830; St. Louis, MO, USA), 1% SDS (Sigma, L3771; St. Louis, MO, USA) and a protease inhibitor cocktail (Sigma, P9599; St. Louis, MO, USA). For each 100 mg of sample in Eppendorf tubes, 1 mL of extraction solution was added, and it was vortexed for 5 min, with 1 min rest intervals. Subsequently, 1 mL of phenol solution (Sigma, P4557; St. Louis, MO, USA) was added in the fume hood to tubes sealed with parafilm to avoid spillage. The tubes were briefly vortexed and placed on ice with shaking for 20 min. Then they were centrifuged at 4 °C and 15,000× *g* for 30 min. The upper phenolic phase was recovered. The volume of each tube was increased to 2 mL with acetone supplemented with 0.07% β-mercaptoethanol (Sigma, M6250; St. Louis, MO, USA), and they were allowed to precipitate overnight at −20 °C. The next day, the tubes were centrifuged at 4 °C and 3000× *g* for 30 min. The supernatant was discarded, and the samples were allowed to dry in the vacuum concentrator. The pellet was resuspended in 300 µL of 1× PBS (Sigma, P5493; St. Louis, MO, USA) supplemented with 1% SDS (Sigma, L3771; St. Louis, MO, USA) by vortexing for 15 min. The tubes were centrifuged at 24 °C and 15,000× *g* for 10 min, and the supernatant was recovered in new tubes. The quantification of the total protein was carried out with the BCA Protein Assay Kit (Thermo Scientific, 23227; Rockford, IL, USA), and the quality of the extract was verified by SDS-PAGE. Samples were stored at −80 °C until use.

### 4.3. Protein Reduction, Alkylation, and Digestion

A total of 100 µg of protein was taken from the previous extract, and the volume was increased to 100 µL with PBS supplemented with 1% SDS solution. For the protein reduction, 10 mM TCEP (Sigma, 68957; St. Louis, MO, USA) was added and incubated for 45 min at 60 °C. Subsequently, the proteins were alkylated for 60 min with 30 mM IAM (Sigma, A3221; St. Louis, MO, USA) in the dark at room temperature. Then 30 mM DTT (Sigma, D9779; St. Louis, MO, USA) was added, and it was incubated for 10 min at room temperature. Cold acetone was added to the tubes and incubated overnight at −20 °C to precipitate the proteins. Next, the tubes were centrifuged at 10,000× *g* for 15 min at 4 °C. The supernatant was discarded, and the pellet was dried in a vacuum concentrator. The dry pellet was resuspended with 50 mM TEAB (Sigma, T7408; St. Louis, MO, USA) supplemented with 0.1% SDS (Sigma, L3771; St. Louis, MO, USA). Finally, the protein content was quantified again with the BCA Protein Assay Kit (Thermo Scientific, 23227; Rockford, IL, USA) and visualized on SDS-PAGE. Samples were stored at −80 °C until use. Proteins were digested with trypsin (Thermo Scientific, 90058; Rockford, IL, USA) 1:30 (trypsin:protein) overnight at 37 °C, followed by incubation with trypsin 1:60 at 37 °C for 4 h. Afterward, samples were vacuum-dried.

### 4.4. Peptide Isobaric Labeling with Tandem Mass Tag (TMT) and Fractionation

The TMT Isobaric Label Reagent Set plus TMT11-131C kit (Thermo Scientific, A34808; Rockford, IL, USA) was used to perform the isobaric labeling. Two biological replicates were used for each tissue. Peptides were labeled with 127C and 128N tags for peptides from leaves, 128C and 129N tags for peptides from callus, and 129C and 130N tags for peptides from suspension cultures. After protein labeling, peptide fractionation was carried out with Pierce™ High pH Reversed-Phase Peptide Fractionation Kit (Thermo Scientific, 84868; Rockford, IL, USA).

### 4.5. Nano LC/MS-MS Analysis

Samples were analyzed by nano LC-MS/MS analysis using an Orbitrap Fusion Tribrid (Thermo Fisher Scientific, San Jose, CA, USA) mass spectrometer equipped with an “EASY spray” nano ion source (Thermo Fisher Scientific, San Jose, CA, USA). The Orbitrap Fusion Tribrid (Thermo Scientific, San Jose, CA, USA) was interfaced with an UltiMate 3000 RSLC system (Dionex, Sunnyvale, CA, USA). Each sample was reconstituted with 0.1% formic acid in LC-MS-grade water (solvent A; Thermo Scientific, 85178; Rockford, IL, USA), and 5 μL was injected into a nanoviper C_18_ trap column (3 µm, 75 µm × 2 cm, Dionex) at 3 μL min^−1^ flow rate, and then separated with a 100 min gradient on an EASY spray C_18_ RSLC column (2 µm, 75 µm × 25 cm), with a flow rate of 300 nL min^−1^, and using solvent A and 0.1% formic acid in 90% acetonitrile (solvent B). The gradient was as follows: 10 min solvent A, 7%–20% solvent B for 25 min, 20% solvent B for 15 min, 20%–25% solvent B for 15 min, 25%–95% solvent B for 20 min, and eight min solvent A. The mass spectrometer was operated in positive ion mode with nanospray voltage set at 2.5 kV and source temperature at 280 °C. External calibrants included caffeine, Met-Arg-Phe-Ala (MRFA), and Ultramark 1621 (88323, Thermo Fisher Scientific^TM^ Pierce^TM^; Rockford, IL, USA).

### 4.6. Synchronous Precursor Selection (SPS)-MS3 for TMT Analysis

Full MS scans were run in the Orbitrap analyzer with 120,000 (FWHM) resolution, scan range 350–1500 *m/z*, AGC of 2.0e5, maximum injection time of 50 ms, intensity threshold 5.0e3, dynamic exclusion one at 70 s, and 10 ppm mass tolerance. For MS2 analysis, the 20 most abundant precursors ions (MS1s) were isolated with charge states set to 2–7. Fragmentation parameters included collision-induced dissociation with collision energy set to 35% and an activation Q of 0.25, an AGC of 1.0e4 with a maximum injection time of 50 ms, a precursor selection mass range of 400–1200 *m/z*, a precursor ion exclusion width of a low of 18 *m/z* and a high of 5 *m/z*, isobaric tag loss TMT and detection run in the ion trap. Afterward, MS3 spectra were acquired as previously described [103] using synchronous precursor selection (SPS) of 10 isolation notches. MS3 precursors were fragmented by HCD with 65% of collision energy and analyzed using the Orbitrap with 60,000 resolution power at 120–500 *m/z* scan range, a two *m/z* isolation window, 1.0e5 AGC, and a maximum injection time of 120 ms with one microscan.

### 4.7. Data Processing

The resulting MS/MS data were processed using the MASCOT (v.2.4.1, Matrix, Science, Boston, MA, USA) search engine implemented in Proteome Discoverer 2.1 (Thermo Fisher Scientific; San Jose, CA, USA). *C. canephora* and viridiplantae Swiss-Prot databases were used for MS/MS data analysis. Search parameters included 10 ppm and 0.6 Da mass tolerance, trypsin digestion with two missed cleavages allowed. Static modifications included cysteine carbamidomethylation, N-terminal TMT6plex, and lysine TMT6plex. Dynamic modifications included methionine oxidation and asparagine/glutamine deamidation. A strict false discovery rate (FDR) was established for peptides and proteins in the respective node of analysis in Protein Discoverer Suite. Differentially abundant proteins were determined as a fold change ≥1.5 for those up-accumulated or ≤0.66 for those down-accumulated, and *p* < 0.05 was used to identify statistical significance. Functional annotation and GO classification of all identified proteins were determined with Blast2GO software against the viridiplantae NCBI Swiss-Prot database, with a default functional annotation pipeline [104]. The heatmaps were generated using the ggplot2 package for R [105]. The InteractiveVenn tool was used to create the Venn diagram [106]. The hierarchical grouping of the KEGG enrichment was carried out with the ShinyGO V0.66 online platform (http://bioinformatics.sdstate.edu/go/; accessed on 6 June 2021).

### 4.8. Identification of Auxin Homeostasis Protein Homologs

The identification of possible homologs of genes that encode proteins involved in the transport and catabolism of auxins was carried out by in silico analysis. The families of genes and proteins selected for this study were: ABCB (ATP-BINDING CASSETTE subfamily B) involved in auxin transport; GRETCHEN-HAGEN 3 (GH3), and IAA LEUCINE RESISTANT1 (ILR1) involved in auxin catabolism. We used each of our *C. arabica* protein sequences to perform a BLAST analysis against The Arabidopsis Information Resource database (TAIR; accessed on 7 September 2021) to determine possible homologs. The sequences with the highest percentage of identity and similarity were considered homologous with Arabidopsis.

### 4.9. Phylogenetic Analysis

For the phylogenetic analysis, the sequences of each protein family were aligned using the “MUSCLE” tool within the MEGA7 software (http://www.megasoftware.net/; accessed on 10 September 2021). The aligned sequences were trimmed for non-aligned residues within regions of more significant variability. The best evolutionary model was determined in each protein family using the tool “Find Best DNA/Protein Models.” Phylogenetic trees were constructed using the maximum likelihood method based on the JTT matrix-based model, with a bootstrap analysis of 100 replicates. A phylogenetic tree was built using the MEGA 7 software (http://www.megasoftware.net/; accessed on 10 September 2021). The sequences of rice were obtained from http://riceplantbiology.msu.edu and NCBI (https://www.ncbi.nlm.nih.gov/; accessed on 9 September 2021). Tomato sequences were obtained from https://solgenomics.net and NCBI, accessed on 9 September 2021. Arabidopsis sequences were obtained from https://www. arabidopsis.org/; accessed on 9 September, 2021 and NCBI; accessed on 9 September 2021. The amino acid sequences of the *C. arabica* proteins identified in this study were used. The accession numbers for each of the sequences used are listed in Appendix A.

## 5. Conclusions

Tissue culture represents an effective method for the conservation, propagation, and genetic improvement of *C. arabica*. Several types of tissues can be used; however, few proteomic studies have been performed to identify the proteins involved in cell differentiation. Auxin plays a fundamental role in the maintenance and development of in vitro plant tissue culture. The most significant difference was found when comparing the proteins accumulated in the suspensions with the plants. The more significant accumulation of proteins, such as some ABCs, GH3.17, UGT75C1, and IBR1, suggests auxin’s control in its active and inactive forms was through the mechanisms of homeostasis given by signaling, transport, conjugation, and hydrolysis as tissue differentiation increases. Our results serve as a baseline characterization of the proteome of three in vitro tissues of *C. arabica* using a TMT-quantitative strategy.

## Figures and Tables

**Figure 1 plants-10-02607-f001:**
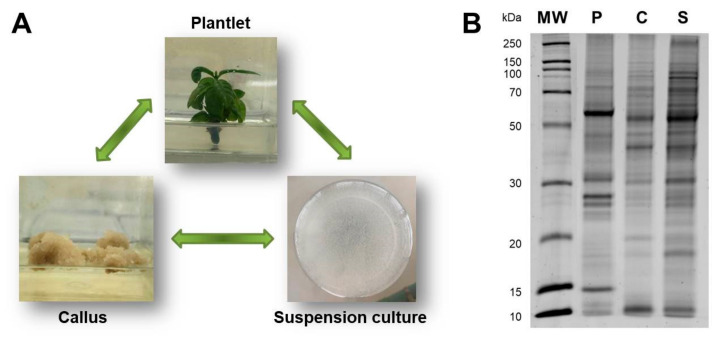
In vitro tissues of *C. arabica*. (**A**) Starting material: plantlet leaves (P), calli (C), and suspension cultures (S); arrows indicate the comparisons that were made between tissues. (**B**) 1D-SDS-PAGE profile of in vitro tissues; the molecular mass standard is indicated on the left side of the gel. Two biological replicates were used for each tissue.

**Figure 2 plants-10-02607-f002:**
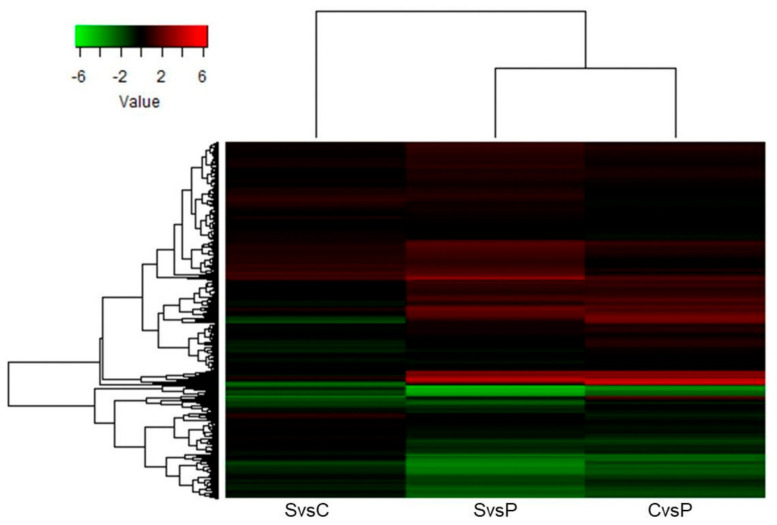
Proteome distribution. Heat map showing the distribution of 2614 proteins among the different tissue comparisons. P: plantlets. C: calli. S: suspension cultures. Down-accumulated proteins are shown in green. Up-accumulated proteins are shown in red.

**Figure 3 plants-10-02607-f003:**
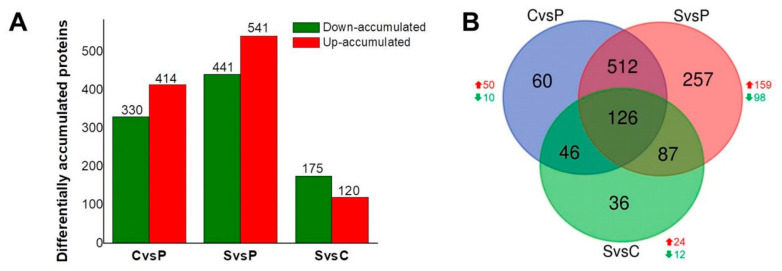
Differentially accumulated proteins. (**A**) The number of differentially accumulated proteins (DAPs) among the different tissue comparisons. Up-accumulated proteins are shown in red >1.5; down-accumulated proteins are shown in green <0.66; *p* < 0.05). (**B**) Venn diagram of the differentially accumulated proteins (DAPs) shared between each comparison. The overlapping regions correspond to the number of shared DAPs. Red and green arrows correspond to the number of up-accumulated and down-accumulated proteins, respectively. P: plantlets. C: calli. S: suspension cultures.

**Figure 4 plants-10-02607-f004:**
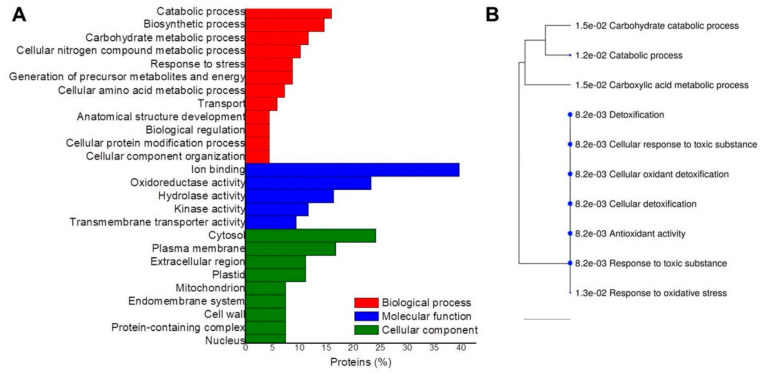
Gene ontology analyses of up-accumulated proteins in the CvsP comparison. The proteins identified when comparing calli vs. plantlets were grouped according to (**A**) GO enrichment and (**B**) hierarchical grouping of the most significant routes based on KEGG.

**Figure 5 plants-10-02607-f005:**
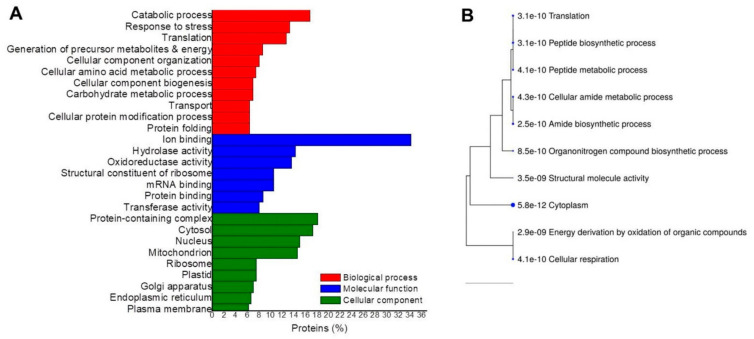
Gene ontology analyses of up-accumulated proteins in the SvsP comparison. The proteins identified when comparing suspension cultures vs. plantlets were grouped according to (**A**) GO enrichment and (**B**) hierarchical grouping of the most significant routes based on KEGG.

**Figure 6 plants-10-02607-f006:**
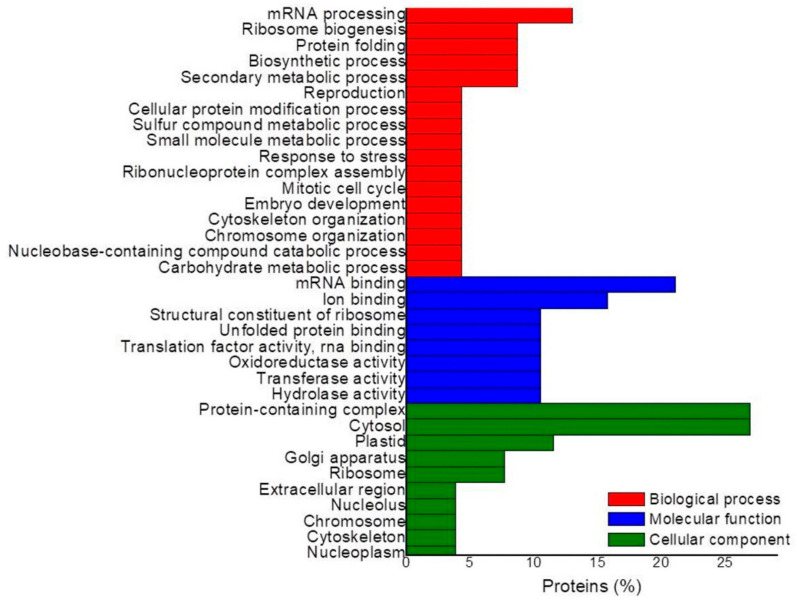
Gene ontology analysis of up-accumulated proteins in the SvsC comparison. The proteins identified when comparing suspension cultures vs. calli were grouped according to GO enrichment. Hierarchical grouping of the most significant routes based on KEGG could not be performed.

**Figure 7 plants-10-02607-f007:**
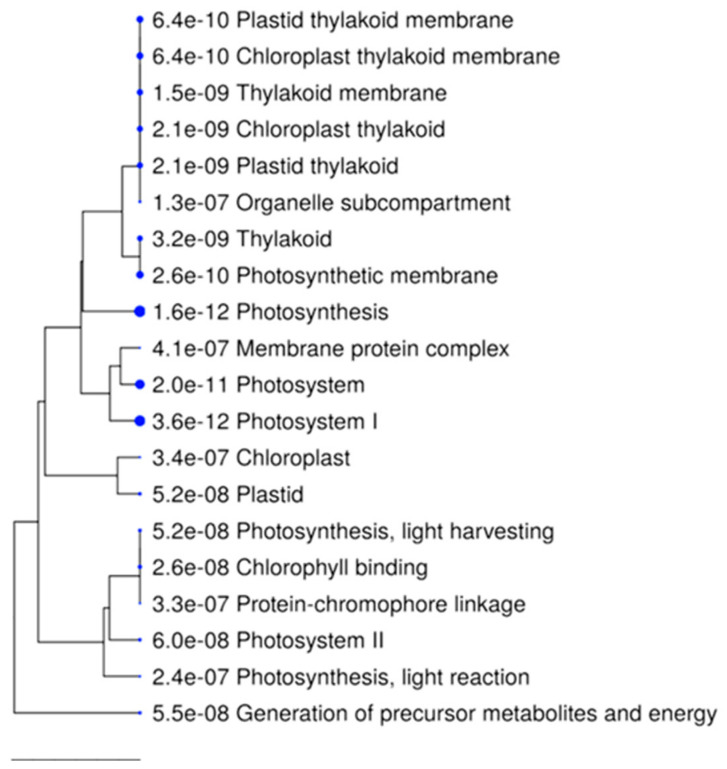
Hierarchical grouping tree of the relationship between significantly enriched routes in constitutive proteins. Clustering of the 126 proteins found consistently across the three tissue comparisons. Pathways with many shared genes are clustered. More prominent points indicate more significant *p* values.

**Figure 8 plants-10-02607-f008:**
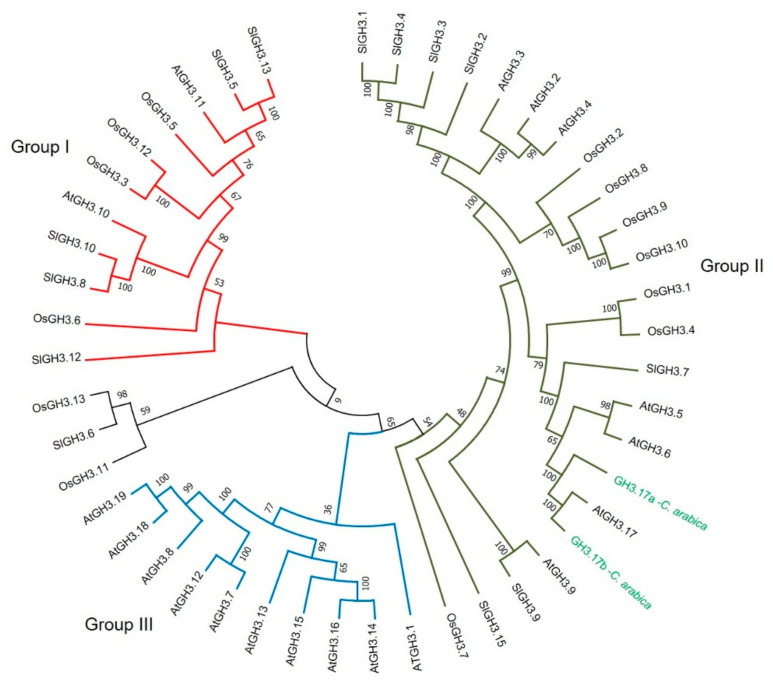
Phylogenetic analysis of the GH3 family involved in auxin conjugation. A phylogenetic tree was constructed to study the phylogenetic relationship of the alignments of 47 GH3 sequences. The red, green, and blue branches represent groups I, II, and III, respectively. GH3 sequences were aligned in MUSCLE. Subsequently, a phylogenetic tree was created using the MEGA7 software. The evolutionary history was inferred using the maximum likelihood method. At: *A. thaliana*, Ca: *C. arabica*, Os: *O. sativa*, Sl: *S. lycopersicum*.

**Figure 9 plants-10-02607-f009:**
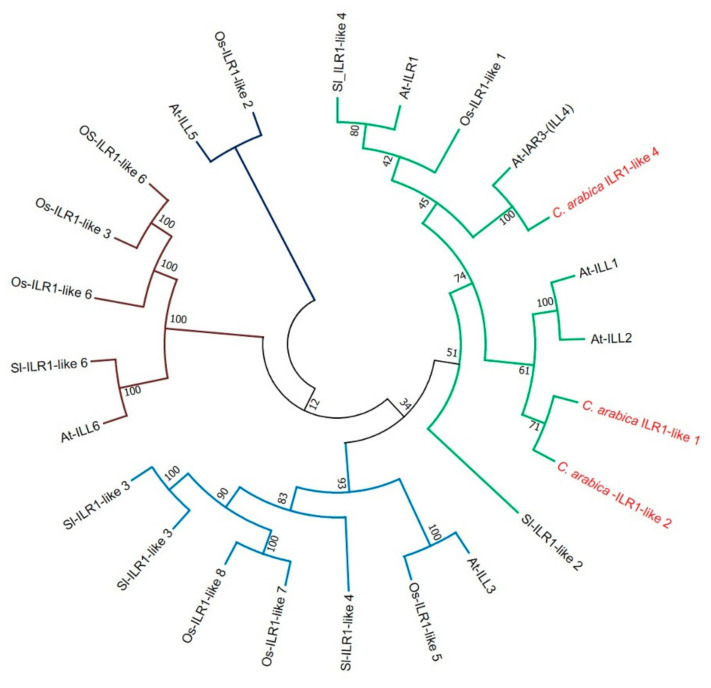
Phylogenetic analysis of the ILR1 family involved in the hydrolysis of auxin conjugates. A phylogenetic tree was constructed to study the phylogenetic relationship of the alignments of 24 ILR1 (ILL) sequences. ILR sequences were aligned in MUSCLE. Subsequently, a phylogenetic tree was created using the MEGA7 software. The evolutionary history was inferred using the maximum likelihood method. At: *A. thaliana*, Ca: *C. arabica*, Os: *O. sativa*, Sl: *S. lycopersicum*.

**Figure 10 plants-10-02607-f010:**
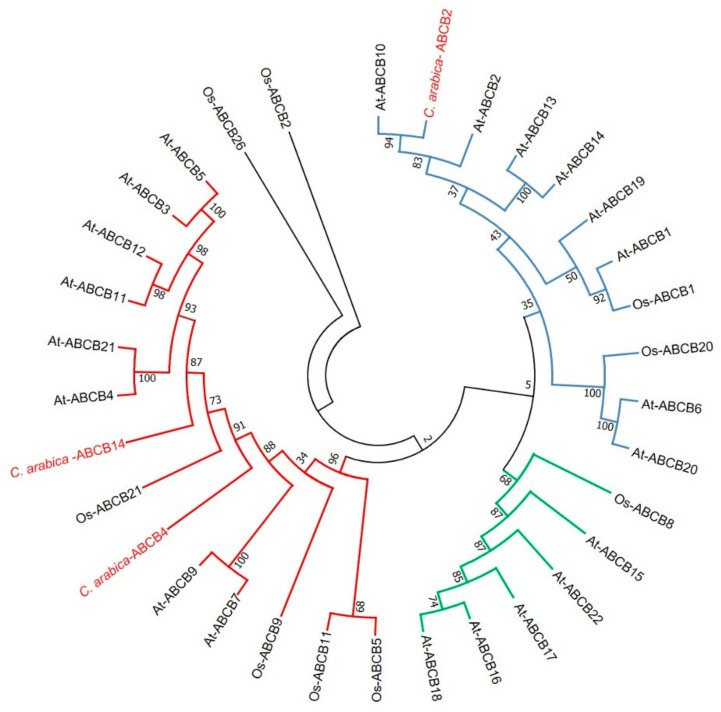
Phylogenetic analysis of the ABCB subfamily involved in auxin transport. A phylogenetic tree was constructed to study the phylogenetic relationship of the alignments of 31 ABCB sequences. The blue, red, and green branches represent the clades I, II, and III, respectively. ABCB sequences were aligned in MUSCLE. Subsequently, a phylogenetic tree was created using the MEGA7 software. The evolutionary history was inferred using the maximum likelihood method. At: *A. thaliana*, Ca: *C. arabica*, Os: *O. sativa*, Sl: *S. lycopersicum.*

**Figure 11 plants-10-02607-f011:**
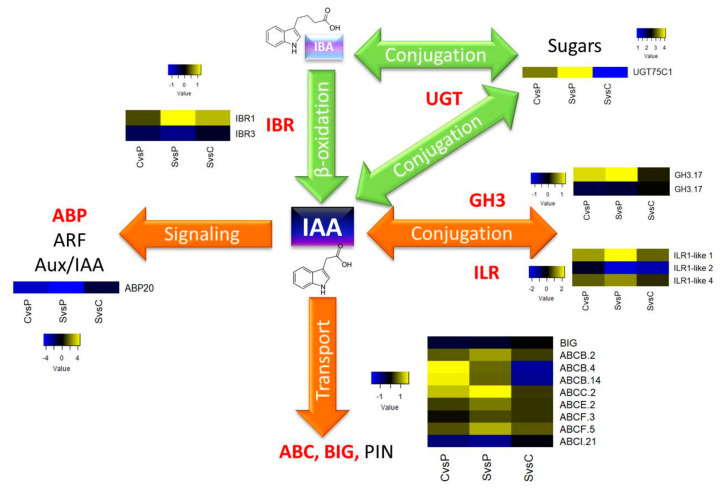
Model of auxin homeostasis during cell differentiation in *C. arabica* in vitro culture. Participation of proteins identified in this study (marked in red) involved in signaling, transport, conjugation, and β-oxidation of auxin.

## Data Availability

Data are available at ProteomeXchange with the identifier. The mass spectrometry proteomics data have been deposited to the ProteomeXchange Consortium via the PRIDE [1] partner repository with the dataset identifier PXD029928.

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
