# Peer review of "Differences in the Abundance of Auxin Homeostasis Proteins Suggest Their Central Roles for In Vitro Tissue Differentiation in Coffea arabica"

_plants, 2021, doi:10.3390/plants10122607_

Round 1

Reviewer 1 Report

Quintana-Escobar et al. report a quantitative proteomics study in Coffea arabica, comparing protein abundance levels in plantlet leaves, calli and suspension culture. The proteomics data is then mined for differences between two types of sample material, with a specific emphasis of proteins involved in auxin homeostasis. This analysis is mainly based on phylogenetics. Finally, the authors provide a hypothesis for regulatory mechanisms of auxin homeostasis in C. arabica.

Overall, I consider this a relevant study because it deals with an organism that is not widely studied in the proteomics community, but is nevertheless of high economic relevance. The manuscript is very well written and easy to follow. The discussion of the results is supported with many figures, which I found helpful.

I am an expert in mass spectrometry-based proteomics, so I consider myself particularly qualified to comment on these aspects of the study. The method employed by the authors (isobaric labeling with TMT and SPS-MS3 analysis) is state-of-the-art and should ensure quantitative data of high quality. With respect to the Materials and methods section, I have a few minor comments:

Line 481: Did the authors really use 3.5 kV voltage for nanospray? This appears to be quite high.
Line 489: "... 20 most abundant MS1s ..." should read "... 20 most abundant precursors ..." (or "precursor ions")
Line 507: arginine deamidation? I assume the authors meant asparagine deamidation?
Line 508: "FDR ... up to 1%" Please specify at which level (peptides, proteins).

I also recommend that the authors deposit their proteomics data in a repository of the ProteomeXchange consortium for reuse by other researchers.

Author Response

Line 481: Did the authors really use 3.5 kV voltage for nanospray? This appears to be quite high.

Answer: dear reviewer, we appreciate this observation. It was a mistake; the parameter used in our study is 2.5 kV

Line 489: "... 20 most abundant MS1s ..." should read "... 20 most abundant precursors ..." (or "precursor ions").

Answer: Thanks for the observation; we changed in the text as ‘precursor ions’

Line 507: arginine deamidation? I assume the authors meant asparagine deamidation?

Answer: Indeed, is asparagine deamidation, thanks for catching this mistake

Line 508: "FDR ... up to 1%" Please specify at which level (peptides, proteins).

Answer: Strict FDR of 0.01 was stablished for peptides and proteins in the respective node of analysis in Protein Discoverer Suite

I also recommend that the authors deposit their proteomics data in a repository of the ProteomeXchange consortium for reuse by other researchers.

Answer: we appreciate your observation. The data is currently being uploaded to the ProteomeXchange consortium. As soon as we have the identifier we will proceed to indicate it in the manuscript.

Reviewer 2 Report

Review

In the manuscript entitled “Differences in the abundance of auxin homeostasis proteins suggest a central role for in vitro tissue differentiation in Coffea arabica” the authors identified proteins involved in auxin homeostasis using baric tandem mass tag (TMT) and the synchronous precursor selection (SPS)-based MS3 technology on three distinct tissue culture material of C. arabica: plantlet leaves, calli, and suspension culture.

The work is well explained and organized. The pictures are of good quality as well.

Major review:

My major concern with the manuscript is in the discussion. Despite the fact the authors identified a large number of proteins differentially accumulated, the discussion is only focused on the proteins involved in auxin homeostasis. And even in this subject, the authors barely associate them to the tissue culture practice and the differences found in the 3 types of biological material used in this study.

I suggest the more tissue culture influence on protein accumulation directed approach and address other types of protein functions other than auxin related ones.

Minor review:

  • Line 29 – Please remove “comparison”
  • Lines 40 – It’s the other way around. It’s not the market that has to meet with the demand but the production.
  • Line 42 – Please add “method” after “preferred”.
  • Line 62 – Please add “dedifferentiated and” before “disorganized”.
  • Line 63-65 – Please, elaborate more and explain better especially when it comes to why tissues with distinct cell types are preferred for SE induction.
  • Line 104 – Remove “each”.
  • Figure 1 – It’s quite confusing having letters (A,B,C) in fig. 1A. Instead of A,B,C, I suggest write “plantlet, calli and suspension cells”.
  • Line 247 – Please review the sentence. The authors state that found in the C. arabica genome proteins… It doesn’t make sense.
  • Line 257-259 – Please, review the whole sentence. It’s difficult to follow the idea.
  • Line 260 – The authors say, “show reduced sensitivity”. To what? Please specify.
  • Figure 11 and from line 383 to 404 – This all section should be in the results section.

Author Response

My major concern with the manuscript is in the discussion. Despite the fact the authors identified a large number of proteins differentially accumulated, the discussion is only focused on the proteins involved in auxin homeostasis. And even in this subject, the authors barely associate them to the tissue culture practice and the differences found in the 3 types of biological material used in this study.

  1. Dear reviewer, we appreciate your contribution and we have responded to this request. We have expanded the discussion with respect to other identified proteins, and to in vitro

I suggest the more tissue culture influence on protein accumulation directed approach and address other types of protein functions other than auxin related ones.

Minor review:

  • Line 29 – Please remove “comparison”

Thank you for the advice. The correction has been made.

  • Lines 40 – It’s the other way around. It’s not the market that has to meet with the demand but the production.

We appreciate the observation. You are right; we have made the change now.

  • Line 42 – Please add “method” after “preferred”.

Thank you for the advice. The correction has been made.

  • Line 62 – Please add “dedifferentiated and” before “disorganized”.

We have completed the sentence. We appreciate the suggest.

  • Line 63-65 – Please, elaborate more and explain better especially when it comes to why tissues with distinct cell types are preferred for SE induction.

Dear reviewer, we have completed the explanation of why leaves are preferred.

  • Line 104 – Remove “each”.
  • Thank you for the suggestion. The correction has been made.

  • Figure 1 – It’s quite confusing having letters (A,B,C) in fig. 1A. Instead of A,B,C, I suggest write “plantlet, calli and suspension cells”.

We greatly appreciate your contribution and have addressed the changes.

  • Line 247 – Please review the sentence. The authors state that found in the C. arabica genome proteins… It doesn’t make sense.

Dear reviewer, we appreciate you have noticed the mistake. We have corrected it since we were referring to the proteome and not the genome

  • Line 257-259 – Please, review the whole sentence. It’s difficult to follow the idea.

We thank you for noticing this detail and we have changed the sentence.

  • Line 260 – The authors say, “show reduced sensitivity”. To what? Please specify.

Thanks for the observation. We have detailed what the sensitivity refers to.

  • Figure 11 and from line 383 to 404 – This all section should be in the results section

We agree with your suggestion, for which we have made the change of section of the figure and its explanation

Reviewer 3 Report

The study was well done and the manuscript was well written. Refer to the Word file for suggested changes.

Author Response

Dear reviewer, we appreciate having noticed the errors. Your corrections have undoubtedly improved the understanding of the manuscript, and we have covered them all.

However, there is one observation that we would like to maintain as is. First of all, we must say that we agree that there is a discussion, sometimes heated, about the terms hormone or phytohormone and plant growth regulator. Without wanting to enter into such a discussion, and based on the evidence published to date, for both hormones and plant growth regulators, we wish to preserve the PGR term. We do not refer to them as hormones for the following reasons: Hormones are synthesized and secreted from glands, and transported to another place where they will perform their function. Hormones do not enter the cell, they use the AMPc to deliver the message into the cell [1].

Auxins were initially named plant growth regulators and also hormones [2]. PGRs are produced in most tissues of the plants. PGRs can act both remotely or locally, that is, being synthesized in one place and then transported to another distant as needed, or they can be synthesized locally by cells, in the site of action [3].

Of course, the topic gives room for a much broader discussion, and at some point, is on the edge of philosophy. However, we believe that the current evidence allows us to speak of growth regulators and not hormones.

References

  1. Robinson, G.A., Butcher, R.W., Sutherland, E.W.  Cyclic AMP. Annu. Rev. Biochem. 1968, 37, 149-174
  2. Bennett, T..;Leyser, O. The auxin question: A philosophical overview. In Zazimalová E., Petrášek J., Benková E. (eds) Auxin and Its Role in Plant Development, Springer: Vienna, 2014; pp. 3-19.
  3. Gaba, V.P. Plant growth regulators in plant tissue culture and development. In Trigiano R.N., Gray D.J. (eds) Plant Development and Biotechnology, CRC Press: Boca Raton, Florida, 2005; pp. 87-98
